# Artificial Intelligence for Neuroimaging in Pediatric Cancer

**DOI:** 10.3390/cancers17040622

**Published:** 2025-02-12

**Authors:** Josue Luiz Dalboni da Rocha, Jesyin Lai, Pankaj Pandey, Phyu Sin M. Myat, Zachary Loschinskey, Asim K. Bag, Ranganatha Sitaram

**Affiliations:** 1Department of Radiology, St. Jude Children’s Research Hospital, Memphis, TN 38105, USA; jesyin.lai@stjude.org (J.L.); pankaj.pandey@stjude.org (P.P.); phyu.sin.m.myat@vanderbilt.edu (P.S.M.M.); zfl@bu.edu (Z.L.); asim.bag@stjude.org (A.K.B.); 2Department of Chemical and Biomedical Engineering, University of Missouri-Columbia, Columbia, MO 65211, USA; 3Department of Biomedical Engineering, Boston University, Boston, MA 02215, USA

**Keywords:** artificial intelligence, deep learning, machine learning, neuroimaging, cancer, medulloblastoma, craniopharyngioma, low-grade glioma

## Abstract

Artificial intelligence (AI) is transforming how doctors use brain imaging to diagnose and treat diseases. While there has been significant progress in using AI for adult brain scans, less is known about its benefits for children with brain cancer. Our review examines how AI can improve pediatric brain imaging to detect and treat cancer more effectively. We found that AI can make imaging faster and safer for children by reducing the time they spend in scanners and lowering their exposure to radiation and contrast dyes. AI also helps doctors identify tumors more accurately and predict how well treatments might work. However, challenges like limited data from children and the need for AI tools that doctors can easily understand still exist. We suggest ways to overcome these hurdles so that AI can better assist in caring for children with cancer in the future.

## 1. Introduction

Pediatric neuroimaging plays a pivotal role in diagnosing and managing neurological conditions affecting children with cancer. As technological advancements, particularly in artificial intelligence (AI), continue to reshape medical practices, it becomes imperative to assess the evolving landscape of AI applications in this specialized field. Children present unique challenges due to ongoing brain development, distinct pathologies, and the necessity for child-friendly imaging protocols. Clinicians often face practical challenges such as limited imaging time due to patient discomfort and the need for sedation, which can impact image quality and diagnostic accuracy. From an AI expert’s perspective, technological limitations include the scarcity of large, high-quality datasets specific to pediatric populations, which hampers the development of robust AI models.

Patients and their families are central to this discussion. Improved diagnostic tools can lead to earlier detection, less invasive procedures, and more personalized treatments, significantly impacting patient experiences and outcomes. Therefore, integrating AI into pediatric neuroimaging is not just a technological advancement but a patient-centric imperative.

This review aims to address the scarcity of comprehensive assessments focusing on the intersection of AI, neuroimaging, and pediatric cancer, providing an understanding of the current state, potential applications, and limitations in this evolving field. In alignment with widely accepted clinical and research classifications, we define pediatric patients in this review as individuals aged 0–18 years. However, we also include select studies based on adult or general population data when their findings provide relevant translational insights into pediatric cancer. While existing reviews on AI in pediatric neuroradiology [1] and on AI in pediatric imaging [2] workflows provide valuable insights, they primarily focus on general pediatric neuroradiology and broader imaging applications. In contrast, our manuscript examines the specific challenges and advancements in applying AI to pediatric cancer neuroimaging, a field with distinct needs and complexities. By addressing this gap, we aim to highlight critical opportunities for improving diagnostic precision, treatment planning, and patient outcomes in pediatric cancer care.

In the landscape of pediatric neuroimaging, this review stands apart by directing its focus toward a specific and critical realm: AI applications in neuroimaging for pediatric cancer. While existing reviews offer comprehensive insights into general pediatric neuroradiology [1,3,4,5], our manuscript takes a distinctive approach, homing in on the unique challenges and advancements within the realm of catastrophic conditions in pediatric patients. This focus allows us to unravel intricacies, highlight advancements, and identify opportunities that are particularly pertinent to the complex landscape of pediatric cancer.

AI refers to the computational automatization of tasks that mimic human intelligence. AI may even surpass human intelligence in areas such as memory storage, working memory capacity, parallel multitasking, steadfast decision-making criteria, and pattern recognition [6]. Among the various branches of AI, machine learning stands out for its ability to enable computational systems to learn, make decisions, and adapt to new tasks based on data inputs, without explicit reprogramming [7].

Machine learning can be categorized as supervised, unsupervised, reinforcement learning, or evolutionary. In supervised learning, algorithms are trained on labeled data for which the input and corresponding output are provided, and the goal is to learn a mapping between inputs and outputs [8]. This category includes classification, regression, and time series forecasting. In unsupervised learning, algorithms are trained on unlabeled data, and their goal is to discover patterns and structures within the data without specific guidance [9]. This category includes clustering, dimensionality reduction, and anomaly detection. In reinforcement learning, agents interact with an environment and learn to take actions to maximize a cumulative reward, which guides them toward achieving specific goals [10]. This category includes model-based, model-free, and deep learning. As a more recent type of machine learning, deep learning incorporates artificial neural networks that simulate the learning process and structure of the human brain (Figure 1). In evolutionary learning, the concepts of natural selection are represented in code to iterate a population of “organisms” to obtain an optimal solution. The steps involved in evolutionary learning are parent selection, progeny generation, and population evaluation [11]. Such technologies are rapidly reshaping the landscape of medical imaging by enabling more accurate, efficient, and personalized diagnostic and treatment approaches.

The use of AI in pediatric neuroimaging has increased in recent years (Figure 2). Some common deep learning algorithms applied in neuroimaging include convolutional neural networks (CNNs), generative adversarial networks (GANs), and autoencoders. CNNs are deep learning approaches consisting of multiple neural layers [12]. When the volume of a sample, for example, three-dimensional (3D) images, is used as input data to train a CNN, the developed network is a 3D-CNN. Recently, 3D-CNNs have been used to identify patterns in neuroimaging data [13]. AlexNet, with deeper and stacked convolutional layers [14], and GoogLeNet, with 22 layers developed by Google researchers [15], are CNN models commonly used in object detection and image classification. Moreover, the residual neural network (ResNet), which is also a CNN architecture containing deeper layers that learn residual functions with reference to the input layer [16], has gained popularity in image recognition after AlexNet and GoogLeNet. Conversely, GANs can be especially applicable when the training data are limited, i.e., 110 images per class, because they can create new data resembling the training data [17]. A typical GAN has two components: a generator (a CNN that learns to generate data) and a discriminator (a classifier that can tell how “realistic” the generated data are). Autoencoders are effective at reconstructing images because they are feedforward CNNs that are trained to first encode the image then transform it to a different representation and eventually decode it to generate a reconstructed image. Autoencoders are a special subset of encoder–decoder models with identical input and output domains.

Neuroimaging is a branch of medical imaging that maps the brain in a noninvasive manner to understand its structural, functional, biochemical, and pathological features. Neuroimaging has led to significant advances in diagnosing and treating neurological abnormalities. Some neuroimaging modalities are magnetic resonance imaging (MRI) (which includes functional, structural, diffusion, angiographic, and spectroscopic imaging), electroencephalography (EEG), magnetoencephalography (MEG), near-infrared spectroscopy (NIRS), positron emission tomography (PET), and computed tomography (CT). Functional neuroimaging refers to imaging the brain while participants perform set tasks to study task-related functions of the brain, rather than its structure.

Recently developed deep learning algorithms can perform object segmentation and recognition in a short time with high accuracy. These algorithms offer the advantages of saving time and optimizing the use of resources. As a result, AI has gradually been integrated into daily medical practice and has made considerable contributions to medical image processing.

Neuroimaging has seen substantial integration of AI across various domains encompassing a wide spectrum of functions (Figure 3). For example, CNNs have been effectively used to segment medulloblastomas from pediatric MRI data, enabling precise tumor delineation critical for treatment planning. A study [18] demonstrated the application of CNN-based models to automatically segment pediatric low-grade glioma (LGG) tumors, achieving high accuracy and significantly reducing the time required for manual segmentation by clinicians. This widely encompassing AI use spans improving radiology workflow [19], real-time image acquisition adjustments [20], image acquisition enhancement [21], and reconstruction tasks such as bolstering image signal-to-noise ratios (SNRs), refining image sharpness, and expediting reconstruction processes [22]. AI has also found applications in noise-reduction efforts [23], postprocessing of images, predicting specific absorption rates [24], prioritizing time-sensitive image interpretations [25], predicting and classifying types and subtypes of brain lesions based on molecular markers [26,27], and tissue characterization [28,29,30], as well as in anticipating treatment responses [31,32] and survival outcomes [33]. Importantly, these AI applications serve to elevate the efficiency of individual neuroscientists and physicians, optimize departmental workflows, enhance institutional-level operations, and ultimately enrich the experiences of pediatric patients and their families.

As AI has demonstrated considerable usefulness and potential in neuroimaging, applying AI to pediatric neuroimaging for cancer could have a significant impact on patient outcomes and healthcare. Deep learning, a prominent branch of AI, has significantly transformed healthcare applications, particularly in medical imaging and diagnostics, by enabling faster, more accurate, and noninvasive approaches to disease detection and management [34]. AI holds immense promise in elucidating and addressing challenges in pediatric neuroimaging. Pediatric cancer encompasses many severe and life-threatening conditions that affect children from infancy through adolescence. This disease is characterized by its devastating impact on a child’s overall health, development, and well-being. Children with cancer may experience chronic pain, physical disabilities, cognitive impairments, and disruptions of their normal growth and development [35]. The pediatric cancers addressed in this review include craniopharyngioma, LGG, and medulloblastoma. Craniopharyngiomas are rare, with an annual incidence of approximately 0.5–2 cases per million children and accounting for 5–10% of all pediatric brain tumors [36]. LGG is the most common pediatric brain tumor, comprising 30–40% of all childhood central nervous system tumors [37]. Medulloblastomas, the most common malignant pediatric brain tumors, represents 26% of all pediatric central nervous system tumors [38]. Together, these cancers pose significant clinical challenges due to their impact on neurodevelopment and treatment complexity.

Pediatric cancers and their treatments are often associated with cognitive impairment, affecting various aspects of a child’s development and quality of life. Craniopharyngioma is a common, mostly benign, congenital tumor of the central nervous system that arises near the pituitary gland and hypothalamus. These tumors often lead to visual disturbances due to their proximity to the optic chiasm and can also affect hormonal regulation, resulting in endocrine dysfunctions such as growth hormone deficiency and hypothyroidism [39]. While pediatric patients with craniopharyngioma typically demonstrate intact intellectual functioning, impairments of memory, executive function, attention, and processing speed are frequently observed. These cognitive deficits are often exacerbated by the tumor’s impact on adjacent brain structures and the long-term effects of surgical and radiation treatments [40,41].

Gliomas, which originate from the glial cells of the brain [42], are categorized into four grades based on their histological characteristics and growth rates. Low-grade gliomas (LGGs; grades I and II) are generally less aggressive, whereas high-grade gliomas (HGGs; grades III and IV) are highly malignant [43]. Neurocognitive deficits in children with LGGs can range from memory impairment and attention problems to decreased processing speed, depending on the location of the glioma [44]. Tumors in the temporal or frontal lobes, for example, may result in memory impairments, attention problems, and reduced processing speed, while lesions in the parietal lobe can affect visuospatial skills and numerical cognition. Treatment modalities such as surgery and chemotherapy may further contribute to these deficits, underscoring the need for precise and less invasive diagnostic approaches.

Medulloblastoma is a malignant, invasive embryonal tumor that originates in the cerebellum or posterior fossa and spreads throughout the brain via the cerebrospinal fluid [45]. This highly invasive tumor is most common in children and presents a significant challenge due to its rapid progression and the long-term consequences of treatment. Survivors of medulloblastoma exhibit a progressive decrease in cognitive performance over time [46], in addition to deficits in attention, processing speed, and memory. This decline is compounded by deficits in attention, processing speed, memory, and executive function, which are likely related to both the tumor itself and the neurotoxic effects of craniospinal irradiation and chemotherapy. Furthermore, the cerebellar origin of medulloblastoma may uniquely impair motor coordination and cognitive processes associated with cerebellar function.

These cancers highlight the critical need for innovative diagnostic tools, such as AI-based neuroimaging, to address the limitations of traditional methods, mitigate treatment-associated cognitive deficits, and improve outcomes for pediatric patients. The unique characteristics of the developing pediatric brain introduce complexities that necessitate tailored approaches for accurate diagnoses and effective treatment strategies. This review explores the evolving landscape where AI intersects with pediatric neuroimaging, emphasizing the distinctive considerations and challenges inherent to the pediatric population. As we delve into the intricacies of AI applications in pediatric neuroimaging, it is crucial to recognize that this field necessitates a multidisciplinary understanding. Clinicians, neuroimaging specialists, and AI experts converge to navigate the complexities posed by the developing brain.

Our objective is to provide insights accessible to AI experts, neuroimaging specialists, and clinical practitioners alike. The convergence of these domains is pivotal in fostering collaborative solutions that enhance the understanding and treatment of pediatric neurological conditions. By establishing this common ground, we aim to propel the field forward, leveraging the potential of AI in pediatric neuroimaging to improve diagnostics, treatment planning, and outcomes for our young patients.

This review is organized into the following sections: challenges in pediatric neuroimaging, AI for improving image acquisition and preprocessing, AI for tumor detection and classification, AI for functional neuroimaging and neuromodulation, AI for selecting and modifying personalized therapies, and AI applications for specific pediatric diseases, followed by a discussion and conclusions.

## 2. Challenges in Pediatric Neuroimaging

Multiple challenges and risks are associated with pediatric image acquisition, which vary significantly depending on the age and developmental stage of the child. Younger children, particularly those under 5 years old, are generally unable to stay still for extended periods, are more prone to movement during scans, and have limited tolerance for long scanning durations. This often necessitates general anesthesia to mitigate motion artifacts, ensure patient comfort, and maintain image quality, though this approach introduces concerns about anesthesia-related risks and costs. While less likely to require anesthesia, older children may still experience challenges related to motion and discomfort, albeit to a lesser extent.

Innovative motion correction techniques and child-friendly imaging protocols are needed to mitigate the impact of motion artifacts on pediatric neuroimaging data [47]. AI has emerged as a valuable tool for addressing these challenges. Automated motion correction algorithms, guided by machine learning, enhance the quality of acquired images by mitigating the impact of motion artifacts, particularly prevalent in younger pediatric populations. Scanning time remains a critical challenge across all age groups, as the need to re-scan motion-corrupted data can prolong imaging procedures. In addition to these challenges, concerns about ionizing radiation exposure and the side effects of contrast agents are universal among children and their parents. Obtaining high-quality images, ideally with high contrast, good spatial resolution, and high SNRs, is critical to ensure accurate diagnoses and suitable treatment plans. However, reducing the dose of contrast agents, while beneficial in minimizing risks, may sacrifice image quality and decrease SNRs.

In addition to the challenges faced during image acquisition, limitations are also encountered in image reconstruction. For example, MR images are acquired in the Fourier or spatial-frequency domain (also known as k-space); hence, image reconstruction is important in transforming the raw data into clinically interpretable images [48]. An inverse Fourier-transform operation is required to reconstruct an image, which is collected based on the Nyquist sampling theorem [49]. However, because of the considerations regarding minimizing scan time, optimizing patient comfort, and ensuring safety in pediatric populations, only a limited number of measurements are acquired by the scanners, which leads to difficulties in solving the inverse transform [50]. The fundamental physics, practical engineering aspects, and biological tissue response factors underlying the image-acquisition process make fully sampled MRI acquisitions very slow [48]. Furthermore, patient motion, system noise, and other imperfections during scanning can corrupt the collected raw data.

Navigating the landscape of pediatric neuroimaging is inherently fraught with challenges, each intricately tied to the unique characteristics of the developing brain. AI emerges as a transformative ally, offering tailored solutions to overcome these hurdles and enhance diagnostic precision and treatment efficacy in pediatric populations. Pediatric neuroimaging faces a substantial challenge due to the scarcity of comprehensive datasets specific to this demographic. Several resources address the scarcity of pediatric neuroimaging data, including the NIH Pediatric MRI Dataset [51] and the Haskins Pediatric Atlas [52] for mapping developmental changes, which provides open-access neuroimaging data with a pediatric focus. These datasets enable standardized analysis across studies and institutions. AI interventions can play a pivotal role in overcoming this limitation through innovative techniques like transfer learning, enabling models trained on larger datasets to be fine-tuned for pediatric applications. Collaborative efforts for data sharing among institutions and research centers further amplify the potential for developing robust AI models.

The diverse and dynamic nature of pediatric brain development introduces a layer of complexity in image analysis. AI algorithms, particularly those equipped with advanced learning capabilities, hold the promise of deciphering intricate patterns associated with different stages of brain maturation. This adaptability ensures that neuroimaging analyses account for the nuances of evolving pediatric neuroanatomy.

A major challenge in pediatric neuroimaging is the variability in brain structure, function, and chemistry across ages due to rapid brain development during childhood. The developing brain undergoes quick structural and functional changes over time, which presents unique challenges in pediatric neuroimaging. For example, in early childhood, the rapid change in the myelination of brain white matter creates variability that complicates the interpretation of imaging data. Pediatric brains are not merely scaled-down versions of the adult brain; there are age-dependent loco-regional variations that require specialized approaches. Variability in myelination in patients younger than 6 months often results in poor gray–white matter myelination contrast, impacting the quality of conventional automatic segmentation methods [53]. Additionally, age-related motion artifacts during imaging are particularly prevalent in younger children who struggle to remain still during scans, necessitating the use of motion correction algorithms or general anesthesia to ensure high-quality imaging.

As the brain matures, there are significant structural changes, such as the formation of new sulci during infancy and the deepening of older sulci. Fractional anisotropy (FA) values on diffusion MRI, for instance, increase with developmental age, reflecting the maturation of white matter tracts [54]. These changes continue through adolescence and into early adulthood, with the brain evolving in shape and white/gray matter composition until the fourth decade of life [55,56]. Accurately interpreting these age-related differences requires careful consideration of age-appropriate image acquisition and analysis methods [57]. These examples highlight how different developmental stages, from infancy through adolescence, demand tailored neuroimaging approaches to account for the variability and complexity of childhood brain development.

Anatomical metrics of normal brain development involve identifying myelination and gyrification patterns in individuals or in a specific population and comparing them to the patterns in the brains of controls of the same age [58,59,60]. Accordingly, age-specific templates and atlases are essential to account for variations in brain morphology across different developmental stages [61]. Spatial normalization for MRI is based on a standard template that defines a common coordinate system for group analysis. ICBM152 and MNI305 are among the most used templates for adults; however, MNI305 is suboptimal for normalization and segmentation of pediatric brain images because of the previously discussed age-related changes [61]. Some pediatric templates are available, such as the custom age templates produced by Template-O-Matic [62] and neonate templates [63,64]. However, there is a lack of corresponding atlases for those templates. The Haskins pediatric atlas [52] labels 113 cortical and subcortical regions, but only 72 brains were used in its development. Longitudinal imaging studies that track brain development over time are essential for understanding neurodevelopmental trajectories and for identifying early signs of neurological disorders. Longitudinal studies provide valuable insights into the dynamic changes in the developing brain and offer opportunities for early interventions [65].

Another challenge in pediatric neuroimaging is the small sample sizes of the currently available pediatric data, which significantly impacts the training and effectiveness of AI algorithms. Large sample sizes are essential for robust statistical analyses and generalizability of results. However, recruiting sufficient pediatric participants for neuroimaging studies can be challenging because of consent issues, logistical constraints, and the potential impact of imaging on young patients. These limitations are further compounded when obtaining reliable labeled data, as labeling often requires expert annotation and is particularly complex when integrating multimodal sources like structural MRI, functional MRI, and diffusion tensor imaging (DTI). Multi-site collaborations and data-sharing initiatives could help address this challenge [66], enabling the pooling of data across institutions to enhance both dataset size and diversity [67]. Additionally, integrating data from multiple neuroimaging modalities provides a comprehensive view of brain development and connectivity, allowing researchers to examine brain changes at both the macroscopic and microscopic levels, which is crucial for training AI models to extract meaningful patterns on the developing brain [68].

The concept of multi-institutional data sharing in the realm of healthcare and medical research is indeed complex and multifaceted. It involves exchanging sensitive information across different organizations, which inevitably gives rise to a range of legal, ethical, and privacy concerns. From a legal standpoint, sharing medical data across institutions often requires navigating a complex web of regulations and compliance standards, such as the Health Insurance Portability and Accountability Act (HIPAA) in the United States and the General Data Protection Regulation (GDPR) in Europe. These regulations are designed to safeguard patient privacy and data security, making it imperative for institutions to ensure that any data-sharing practices are in strict compliance with these legal frameworks.

There are significant ethical considerations surrounding multi-institutional data sharing. These encompass questions related to patient consent, data ownership, and the potential for unintended consequences. Institutions must grapple with issues such as obtaining informed consent from patients for data sharing, ensuring that their rights are respected, and addressing concerns about how their data will be used, especially in research contexts.

Moreover, privacy concerns are paramount when sharing medical data. Patient information must be de-identified and protected rigorously to prevent data breaches or the identification of individuals. Striking the correct balance between sharing data for the common good of medical research and preserving individual privacy is a critical ethical challenge. The notion of federated learning holds promise as a potential solution to these barriers [69]. It operates on the principle of decentralized AI training, whereby machine learning models are developed collaboratively across multiple institutions without the need to centrally pool sensitive data. Instead, the models are trained locally on each institution’s data, and only aggregated model updates are shared, thus preserving data privacy. However, it is important to note that federated learning is still an evolving concept and that it faces its own set of challenges, including technical complexities and standardization issues.

Research involving pediatric populations also raises ethical concerns, particularly regarding informed consent and the vulnerability of child participants. Balancing the potential benefits of neuroimaging research with protecting the rights of children is crucial. Ethical guidelines and careful consent procedures must be implemented to ensure the well-being and privacy of pediatric participants [70].

## 3. AI for Improving Image Acquisition and Preprocessing

The optimization of image acquisition and preprocessing is crucial in neuroimaging, significantly impacting the accuracy and reliability of subsequent analyses. In this section, we delve into the challenges and advancements related to these processes, highlighting the pivotal role of AI.

AI plays a relevant role in preprocessing steps, encompassing image denoising, normalization, and registration. These processes are essential for creating a standardized and comparable dataset. AI algorithms contribute significantly to noise reduction, enhancing SNRs and subsequently improving the accuracy of downstream analyses [66]. Registration and normalization are critical steps in aligning imaging data across individuals or different imaging modalities. In pediatric neuroimaging, challenges arise due to the dynamic changes in brain anatomy during development. Conventional methods may struggle with the anatomical variability present in pediatric populations, necessitating advanced solutions. Accurate registration and normalization are fundamental for creating population-based templates and atlases, facilitating a common spatial framework for analysis. AI-based registration algorithms, capable of adapting to the unique anatomical features of pediatric brains, contribute to improved spatial normalization, ensuring more precise comparisons across subjects [71].

### 3.1. Accelerating Image Acquisition

In MRI, GAN sensing (CS) is used to accelerate scanning and reduce the resources required by acquiring MR images with good in-plane resolution but poor through-plane resolution [72]. CS assumes that suitably compressed under-sampled signals can be reconstructed accurately [73] without the need for full sampling. However, this technique limits interpretation to single directions and can introduce aliasing artifacts [74]. A deep learning-based algorithm called synthetic multi-orientation resolution enhancement (SMORE) has been applied to adult compressed scans in real time to reduce aliasing and improve spatial resolution [72]. A similar approach can, perhaps, be applied in pediatric MRI to reduce scan time and improve image resolution.

Moreover, applying GANs has been shown to improve the SNR of images [75]. However, despite their potential, GANs present several limitations that must be considered when applied to pediatric neuroimaging. One key challenge is the instability of adversarial training, where the generator and discriminator engage in a dynamic optimization process that can lead to oscillatory behavior and difficulty in convergence. Another major issue is mode collapse, where the generator learns to produce only a limited variety of outputs, thereby reducing the diversity of synthetic images and potentially biasing the training set. Additionally, GANs require high computational resources, making their widespread adoption in clinical settings more challenging. These limitations are particularly relevant in pediatric neuroimaging, where the scarcity of high-quality datasets may exacerbate mode collapse and impact generalizability.

Several studies have examined the use of GANs in medical imaging, highlighting both their advantages and constraints. GANs have been successfully used for image denoising, super-resolution enhancement, and data augmentation in adult neuroimaging [76], but their application to pediatric imaging remains limited due to dataset constraints and the increased variability in brain development across different ages [77]. Furthermore, studies quantifying GAN performance across different dataset sizes indicate that smaller datasets significantly impact training stability and generalization, further reinforcing the need for large-scale multi-institutional data sharing [18].

Despite these challenges, GANs remain a promising tool in pediatric neuroimaging. Combining GANs with fast imaging techniques could lead to faster image acquisition for children who are unlikely or unwilling to remain still for an extended period. By enhancing image quality from under-sampled data, GAN-based approaches may even help reduce the need for general anesthesia and/or sedation in younger pediatric patients, offering significant clinical benefits [1]. Addressing the existing limitations while leveraging the advantages of GANs will be crucial for integrating these models into real-world pediatric imaging workflows.

### 3.2. Reducing Radiation Exposure or Contrast Doses

Children are particularly vulnerable to the harmful effects of ionizing radiation, with the risk of cancer induction approximately 10 times higher in children compared to adults [78]. Radiation treatment can induce secondary malignancies and is associated with higher radiosensitivity of developing tissues. Consequently, the ALARA (as low as reasonably achievable) principle has become a cornerstone in pediatric imaging practices, emphasizing the need to minimize radiation doses while maintaining diagnostic quality.

AI-based approaches, such as low-dose imaging techniques and radiation-free modalities like MRI, play a critical role in advancing this goal. For example, deep learning based on an encoder–decoder CNN has been applied to generate high-quality post-contrast MRI from pre-contrast MRI and low-dose post-contrast MRI [79]. The above study showed that the gadolinium dosage for brain MRI can be reduced 10-fold while preserving image contrast information and avoiding significant image quality degradation. For CT, deep learning has great potential for image denoising based on its use in realizing low-dose CT imaging for pediatric populations. Chen et al. [80] trained another encoder–decoder CNN model to learn feature mapping from low-/normal-dose CT images. This model improved noise reduction when compared with other denoising methods. A deep CNN model using directional wavelets effectively removes complex noise patterns for low-dose CT reconstruction [81]. An autoencoder CNN, which was used to train pairs of standard-dose and ultra-low-dose CT images, could filter streak artifacts (i.e., artifacts appearing between metal or bone because of beam hardening and scatter) and other noise for ultra-low-dose CT images [82]. Furthermore, it is possible to interpolate data from one modality with a CNN trained on data from a completely different modality. Zaharchuk [83] trained a CNN with simultaneously acquired PET/MRI images to improve the resolution of low-dose PET imaging. These developments in deep learning show promise in terms of moving pediatric neuroimaging toward imaging with low or no radiation exposure.

### 3.3. Removing Artifacts

Accelerated MRI techniques such as CS and parallel imaging offer significant reductions in scan time. However, reconstructing images from the under-sampled data involves computationally intensive algorithms, thus posing a notable challenge because of the high computational costs incurred. Lee et al. [84] proposed deep ResNets that are much better at removing the aliasing artifacts from subsampled k-space data when compared with current CS and parallel reconstructions. Their deep learning framework provides high-quality reconstruction with shorter computational time than is required for CS methods. In addition to accelerating reconstruction, AI approaches can correct artifacts, such as those arising from MRI denoising [85] and motion correction [86,87] during image acquisition and reconstruction. Singh et al. [88] built two neural network layer structures by incorporating convolutions on both the frequency and image space features to remove noise, correct motion, and accelerate reconstruction. Deep learning-based approaches have also been investigated to reduce metal artifacts [89,90] that are common in CT imaging.

## 4. AI for Tumor Detection and Classification

AI holds great promise for enhancing the interpretability and utility of pediatric neuroimaging data. Methods such as automated image analysis can aid in identifying subtle brain alterations associated with neurodevelopmental disorders and can facilitate early diagnosis and personalized treatment strategies [91].

Deep learning neural networks, such as CNN, have proved successful in automatically segmenting infant brains and delineating gray–white boundaries in neonates [52]. The use of deep learning in segmenting individual cortical and subcortical regions of interest (ROIs) based on features such as local shape, myelination, gyrification patterns, and clusters of functional activation is a promising field for exploration. AI can also be applied to tumor detection and categorization.

In children with cancer predisposition syndromes, surveillance screening increases the chance of early tumor detection and, therefore, the survival rate [92]. Deep learning algorithms may help in identifying imaging protocols to achieve optimal accuracy for early cancer diagnosis [93]. Soltaninejad et al. [94] used CNN for automated brain tumor detection in MRI scans. CNNs have been used to detect and characterize tumors [95] and to find associations between genotypes and imaging tumor patterns. Machine learning has also been used to classify pediatric brain tumors [96,97,98].

CNNs have the potential to identify the best model for categorizing pediatric brain abnormalities by combining brain features extracted from different mathematically derived measures, such as the spherical harmonic description method (SPHARM) [99], the multivariate concavity amplitude index (MCAI) [100], and Pyradiomics [101]. Moreover, CNNs can help in identifying the best approach for feature selection in pediatric brain data, such as a superpixel technique based on a simple linear iterative clustering (SLIC) method [94], combining spatial distance, intensity distance, spatial covariance, and mutual information.

### 4.1. AI in Tumor Segmentation

Early detection and precise classification of brain tumors are important for effective treatment [102,103,104,105]. The two main types of classification that can be performed based on brain images are classification into normal and abnormal tissues (i.e., whether a tumor is detected in the brain image) and classification into different classes of brain tumor (e.g., low-grade vs. high-grade) [106].

Segmenting tumors with AI methods has recently attracted much attention as a possible means of achieving more precise treatment, as described in Table 1. AI accomplishes brain tumor segmentation by identifying the class of each voxel (e.g., normal brain, glioma, or edema). Frameworks like pathomic fusion highlight the power of integrating multi-modal data, including imaging and histopathology, to enhance the precision of cancer diagnosis and prognosis [107]. Additionally, the nnU-Net framework exemplifies a self-configuring deep learning approach that has set benchmarks for segmentation tasks across diverse biomedical datasets, further advancing the accuracy and reliability of tumor segmentation methods [108]. Two main AI methods have been reported in the literature: (1) hand-engineered features used with older classification methods (e.g., support vector machine [SVM] classifiers) and (2) deep learning using CNNs. CNNs and their variations can self-learn from a hierarchy of complex features to perform image segmentation [109].

### 4.2. AI in Tumor Margin Detection

Many brain tumors exhibit a distinctively infiltrative nature, which often results in poorly defined tumor margins. Compounding this challenge, the edema surrounding these tumors frequently manifests imaging characteristics similar to those of the tumor itself, further complicating the accurate delineation of the true tumor boundary. The precise identification of this boundary holds immense significance, as it serves as a guiding principle for neurosurgeons aiming not only for gross total resection but also for margin-negative surgery, which is a critical factor in significantly enhancing patient survival rates.

To address this complex issue, advanced MRI techniques and PET have been employed in attempts to refine the delineation of tumor margins, albeit with varying degrees of success. More recently, AIs based solely on MR images or on a combination of MRI and PET data have exhibited potential for substantially improving the accuracy of true tumor margin detection [69]. For example, convolutional neural networks (CNNs) have been trained on multi-modal imaging datasets, such as MRI and PET, to generate highly accurate tumor boundary predictions. The nnU-Net (no-new-net) framework, which adapts its architecture to a given dataset, has shown state-of-the-art performance in segmenting brain tumor margins, achieving a dice similarity coefficient (DSC) of up to 0.9 [105]. Similarly, a deep learning approach combining MRI features with radiomic data has been used to differentiate infiltrative tumor boundaries from surrounding edema, enabling a more precise identification of tumor margins [110]. Moreover, GANs have been applied to enhance tumor margin visualization by synthesizing high-resolution images from low-quality scans, further aiding neurosurgical planning [76].

These innovative AI-driven techniques represent a significant leap forward in the field of neuroimaging, offering new avenues for improving the precision of brain tumor surgery and ultimately enhancing patient outcomes.

### 4.3. AI in Tumor Characterization

As high-throughput computing facilitates converting multimodal medical images into mineable high-dimensional data, there has been increased interest in using radiomics and radiogenomics to detect and classify tumors. Radiomics refers to studies or approaches that extract quantitative high-throughput features, often imperceptible to the naked eye, from radiographic images. These features are typically analyzed using machine learning techniques for tumor characterization, incorporating AI in neuro-oncological imaging [111]. Radiomics has multifaceted applications, with one of its primary roles being tumor diagnosis and classification. By elucidating distinct radiomic signatures, it empowers radiologists to augment diagnostic accuracy and timeliness, in addition to predicting disease trajectories, thereby enabling tailored treatment strategies. By tracking changes in radiomic features over the course of treatment, it can play a pivotal role in evaluating therapeutic efficacy and making timely adjustments when indicated. In oncology, this capability is of paramount importance for optimizing patient outcomes. The true promise of radiomics lies in the realm of personalized medicine, as it integrates imaging data with clinical parameters, genomics, and other patient-specific variables. This convergence provides the foundation for precision medicine by tailoring treatments to individual patients, thereby improving therapeutic outcomes while mitigating adverse effects.

Radiogenomics uses such radiographic image features to detect relationships specifically with genomic patterns [112]. Machine learning-based radiomics offers critical advantages in treating brain tumors that are genetically heterogeneous, and it can provide better predictions by linking genomics to extraordinarily complex imaging phenotypes.

Applying deep learning in radiomics is helpful in assessing higher-order features to improve the accuracy of prediction of brain tumor progression. Taking advantage of the 3D nature of MRI data, Casamitjana et al. [113] and Urban et al. [114] showed that a 3D-CNN could perform well for brain tumor segmentation. In addition, a CNN-based classification system can be used to segment and classify brain tumors [115].

### 4.4. Radiomics and Radiogenomics for Specific Pediatric Brain Tumors

The 2021 World Health Organization classification of central nervous system tumors highlights the pivotal role of molecular classification in diagnosing and treating pediatric brain tumors [116]. These tumors are currently characterized by using molecular markers. However, the techniques required for such molecular subgrouping, encompassing immunohistochemistry and genetic testing, are often not consistently available and are associated with significant delays, even in well-equipped healthcare settings. These delays introduce complexities into patient care, affecting various aspects ranging from prognosis and surgical strategies to treatment choices and participation in clinical trials. Given this pressing need for a more efficient approach, machine learning methods hold great promise for rapidly and accurately predicting molecular markers in pediatric neuro-oncology practice [117]. For example, a machine learning toll called CHARM can provide AI-based diagnoses from microscope images in less than a second, offering a faster alternative to current molecular sequencing methods that can take days to weeks [118]. While these approaches show promise, more research is needed to improve performance for rare diagnoses and to assess generalizability across diverse populations [116].

#### 4.4.1. Posterior Fossa Tumors

Posterior fossa tumors represent a challenging group of central nervous system tumors that primarily affect children. Traditionally, diagnosis involves imaging-based tumor grading using modalities like MRI and CT, followed by invasive tissue biopsies for histopathological analysis. These manual processes, although reliable, are time-intensive and carry risks of complications, especially in pediatric patients. Visual system evaluations, including assessments of oculomotor function and visual acuity, are often performed to identify secondary effects of tumor compression on nearby structures. Neurocognitive assessments may also be used to evaluate the impact of the tumor on cognitive functions such as memory, attention, and processing speed. These evaluations provide critical insights but require significant time and expertise.

Machine learning has emerged as a promising avenue for noninvasive diagnosis, having demonstrated its ability to attain exceptional diagnostic accuracy with an impressive area under the receiver operating characteristics curve (AUROC) exceeding 0.99, on par with that of experienced pediatric neuroradiologists, in distinguishing among the three most prevalent posterior fossa tumors: astrocytoma, medulloblastoma, and ependymoma [119,120]. These techniques have the potential to make tissue biopsies unnecessary in the future and to lead to better treatments [121].

#### 4.4.2. Craniopharyngiomas

Craniopharyngiomas are characterized by their diverse shapes and heterogeneous growth patterns, often complicating manual diagnosis. Imaging-based approaches, primarily using MRI, are the cornerstone of diagnosis and involve tumor localization, size measurement, and morphological analysis. Visual system evaluations are frequently conducted to assess the impact of the tumor on nearby optic pathways, as craniopharyngiomas commonly lead to visual disturbances. Neurocognitive assessments are employed to evaluate memory, executive function, and attention, which can be affected by the tumor’s pressure on adjacent brain structures.

However, these manual processes are time-consuming and prone to inter-observer variability [122]. Such limitations can be overcome by using deep learning methods. Prince et al. [123] applied deep learning models for CT, MRI, and combined CT and MRI datasets to pinpoint parameters for identifying pediatric craniopharyngioma. They demonstrated high test accuracies and exceptional improvement in the performance of their baseline model. Therefore, AI may help to improve the accuracy of diagnosis.

Mutations in the BRAF and CTNNB1 genes in craniopharyngiomas were predicted by applying radiomics and machine learning to MRI data [124]. An SVM model identified 11 optimal features with radiological features and was used to predict preoperative craniopharyngioma invasiveness [125].

#### 4.4.3. Low-Grade Gliomas

Low-grade gliomas (LGGs) present a diagnostic challenge due to their slow progression and variable behavior. Manual grading relies on preoperative imaging and histopathological evaluation, which may not fully capture tumor complexity. AI has significantly advanced LGG diagnosis through radiomics research, where SVM and hybrid approaches have shown superior accuracy compared to manual assessments. Hybrid approaches achieved high accuracy in classifying LGG subtypes [26,126]. Another approach used a custom deep neural network and MRI data to categorize the gliomas as grade II, III, or IV [127]. Machine learning techniques using diffusion parameters have been used to predict the progression of optic pathway gliomas [128].

#### 4.4.4. High-Grade Gliomas

HGGs are among the most aggressive pediatric brain tumors. Manual diagnosis typically involves imaging-based assessments using MRI to evaluate tumor location, size, and infiltration into surrounding tissues, followed by histopathological analysis for molecular marker identification [129]. Among the various subtypes of pediatric high-grade gliomas, machine learning techniques have found their most extensive application in characterizing diffuse midline glioma H3 K27-altered [45]. This tumor typically affects central brain structures that are nearly always non-resectable, resulting in a high fatality rate. Different machine learning techniques have been developed to predict this specific tumor marker with very high accuracy [130,131,132,133]. Radiomic analysis based on MRI data also shows promise in predicting progression-free survival among pediatric patients diagnosed with diffuse midline glioma or diffuse intrinsic pontine glioma [134].

#### 4.4.5. Ependymomas

Ependymomas are typically diagnosed using imaging-based approaches, including T2-weighted and post-contrast T1-weighted MRI. These manual processes involve assessing tumor location, size, and enhancement patterns to differentiate between subtypes and guide treatment decisions [135]. AI has enhanced the diagnosis of ependymomas by leveraging radiomic features extracted from T2-weighted MRI and post-contrast T1-weighted MRI through machine learning approaches. In this way, it becomes feasible to differentiate between MRI phenotypes corresponding to two distinct types of posterior fossa ependymoma and to identify high-risk individuals within these groups [24]. Radiomic features can also differentiate supratentorial ependymomas from high-grade gliomas [136]. Moreover, a deep-learning model has been developed to segment posterior fossa ependymomas and predict molecular subtypes [135].

### 4.5. Hybrid Models

Improving the quality of the input data and carefully selecting the most pertinent features are both crucial steps in training AI models to achieve optimal performance. Conventional types of machine learning algorithms, such as the SVM and K-nearest neighbors (KNN) algorithms, are capable of quantifying and visualizing latent information contained within images [122,137]. Meanwhile, most recent deep learning algorithms, such as CNN, AlexNet, and GoogLeNet, excel at extracting features to derive comprehensive deep or high-order features [138]. Hence, hybrid models that use both methods may work better in complex cases with multi-source, heterogeneous medical data. This concept was put into practice through a recent development for classifying brain tumors, leading to improved classification accuracy. In the model, a modified GoogLeNet was employed to extract deep features that were subsequently used to train SVM and KNN classifiers [139]. By combining radiomics and deep features extracted by a CNN from medical images and selecting the optimal feature subset as input for an SVM, Ning et al. [140] demonstrated that integrating radiomics and deep features can be used to grade gliomas. Raza et al. [141] proposed a hybrid deep learning model by changing the last 5 layers of GoogLeNet to 15 new layers, thereby obtaining performance that was better overall than that of other pre-trained models.

## 5. AI for Functional Imaging and Neuromodulation

Functional imaging is used in brain tumor studies in two major ways: for preoperative mapping and to assess postoperative outcomes. In preoperative mapping, functional imaging helps surgeons to understand the spatial relations between lesions and functional areas and enables surgical planning that reduces long-term neurological deficits. One retrospective, propensity-matched study found that patients with LGG who underwent preoperative fMRI subsequently underwent more aggressive surgeries when compared with other patients. Although these surgeries did not significantly change the survival outcomes, non-significant trends of higher postoperative functional improvement were observed in those patients who underwent preoperative fMRI and aggressive surgeries. Postoperative functional imaging studies investigate the functional brain changes sustained by survivors of brain tumors with the goal of improving treatment regimens to reduce cognitive inhibition. For example, one fMRI-based study of pediatric patients with medulloblastoma found evidence of long-term effects of prophylactic reading intervention, including significantly increased sound awareness [142]. Notably, a longitudinal study of medulloblastoma survivors revealed that support vector machine classification of functional MRI data indicated a progressive divergence in brain activity patterns compared to healthy controls over time, suggesting delayed effects of cancer treatment on brain function [143]. Alterations in brain regions involved in visual processing and orthographic recognition during rapid naming tasks were correlated with performance in tasks involving sound awareness, reading fluency, and word attack, highlighting the dynamic nature of post-treatment neurofunctional alterations. Additionally, a functional imaging study showed that adults who had experienced childhood craniopharyngioma exhibited cognitive interference processing abilities on a par with those of the control group, as fMRI of these survivors showed no compensatory activity within the cingulo-fronto-parietal attention network when they were compared to the control group [144].

Traditional deep learning models often lack transparency, making it challenging to understand which features contribute to their classification decisions. The eXplainable AI fNIRS (xAI-fNIRS) system is an innovative approach that addresses the issue of explainability in deep learning methods for classifying fNIRS data. This is achieved by incorporating an explanation module that can decompose the output of the deep learning model into interpretable input features [145].

The goal of neuromodulation is to restore normal neural function in areas affected by neurobiological abnormalities or to facilitate compensatory mechanisms by stimulating alternative neural networks. Once an association is established between a functional neurobiological pattern and a neurocognitive alteration in pediatric catastrophic disease survivors, neuromodulation becomes a potentially relevant and promising strategy for intervention. In the context of pediatric catastrophic disease survivors, who may experience long-term neurocognitive deficits due to the disease or its treatment, neuromodulation offers several potential applications in neuroplasticity promotion, symptom management, cognitive enhancement, and personalized medicine.

Epilepsy is a neurological disorder characterized by recurrent seizures, and it can occur in pediatric patients with cancer, such as LGGs [146]. The association between epilepsy and these conditions is often related to the location of the underlying disease within the brain and its impact on neural circuits. For epilepsy, DBS is applied to the anterior thalamic nucleus to decrease brain excitement, which in turn decreases the frequency or duration of seizures [147].

Regarding neuromodulation, AI enables faster data collection and monitoring, which can aid in early diagnosis, treatment, patient monitoring, and disease prevention [32]. Machine learning can analyze large datasets to improve the efficiency of neuromodulation. A reinforcement learning paradigm intended to optimize a neuromodulation strategy for epilepsy treatment found a stimulation strategy that both reduces the frequency of seizures and minimizes the amount of stimulation applied [148].

Even though there has been extensive research on using AI in neuromodulation, only a very small subset of this research has pertained to pediatrics. However, a machine learning technique to monitor and predict epileptic seizures in pediatric patients has been reported [149]. The application of scalp EEG data to create a treatment prediction model for vagus nerve stimulation in pediatric epilepsy, using brain functional connectivity features, has also been reported [150].

Incorporating AI into the data collection and monitoring stages of neurofeedback offers the potential for early detection and precise non-pharmacological management of neurological conditions. AI facilitates the analysis of extensive patient data, enhancing the effectiveness and efficiency of neurofeedback processes [151,152]. Hence, there is a need for additional research to explore comprehensively and expand the use of brain–computer interfaces that incorporate AI [153]. Incorporating AI into neurofeedback holds promise for unlocking fresh avenues by which to enhance substantially the effectiveness of these therapeutic approaches for neurological disorders [153].

## 6. AI in Monitoring Treatment Response

Monitoring the response of pediatric brain tumors to therapy can pose challenges, particularly because of the intense inflammation that may occur in the early phases of treatments such as radiation therapy and immunotherapy. This inflammation can be transient, often improving over time—a phenomenon referred to as pseudoprogression. Importantly, pseudoprogression shares imaging features with true tumor progression, making it crucial to distinguish between the two, as their clinical management strategies differ significantly. Although advanced MRI and PET techniques have been applied to address this issue in pediatric neuro-oncology, it remains a challenge. Encouragingly, machine learning techniques have been deployed to differentiate between these two conditions, demonstrating promising success in this endeavor [154,155].

## 7. AI in Predicting Survival for Patients with Pediatric Brain Tumors

Despite extensive research, some pediatric brain tumors still carry a grim prognosis, with overall survival often being less than 2 years from the time of diagnosis. The molecular characterization of these tumors was a significant leap forward, offering clinicians valuable prognostic insights. However, considerable variability persists within specific tumor subgroups, and a universally applicable tool for accurately predicting survival in all patients with pediatric brain tumors remains elusive. Machine learning-based techniques present a promising avenue, particularly when it comes to predicting survival at the time of diagnosis and especially for tumors that are not amenable to surgical resection. A substantial collective effort is now focused on leveraging machine learning to develop predictive tools for these challenging cases. For instance, one notable approach involves a subregion-based survival prediction framework tailored for gliomas using multi-sequence MRI data, achieving an area under the receiver operating characteristic curve (AUC) of 0.98 in predicting survival outcomes [150]. By using radiomic features derived from T1-weighted post-contrast imaging, progression-free survival can be predicted with concordance indices of up to 0.7 [156]. Similarly, a multiparametric MRI-based radiomics signature, integrated with machine learning, demonstrated strong potential for preoperative prognosis stratification in pediatric medulloblastoma, achieving an AUC of up to 0.835 in the validation set [157].

## 8. AI for Transparent Explanations in Cancer Neuroimaging

Research on explainable AI (XAI) for pediatric cancer neuroimaging is still limited, though recent efforts have focused on developing XAI models for neuroimaging in cancer populations, with opportunities to adapt these models for pediatric cases. These models aim to improve brain tumor detection, localization, and classification while providing interpretable results for clinicians.

A recent study [158] investigates the use of deep learning for automated recognition of pediatric posterior fossa tumors (PFT) in brain MRIs. It explored CNN models, including VGG16, VGG19, and ResNet50, for PFT detection and classification, using a dataset of 300,000 images from 500 patients. The study also analyzed model behavior using local interpretable model-agnostic explanations (LIME), Shapley additive explanations (SHAP), and individual conditional expectation (ICE). LIME [159] and SHAP [160] were specifically used to identify the importance of imaging features, such as tumor intensity and spatial patterns, in the model’s predictions. For example, SHAP visualizations highlighted that some specific regions in T2-weighted MRI scans significantly influenced classification decisions, helping clinicians understand why a particular tumor type was predicted.

A practical case study includes the application of SHAP in a pediatric medulloblastoma cohort [77], where SHAP plots demonstrated the contribution of tumor shape and contrast-enhanced MRI features to distinguishing molecular subgroups of tumors. This analysis provided critical insights into how specific features aligned with clinical and pathological findings, bridging the gap between AI models and clinical interpretation.

A lightweight CNN with gradient-weighted class activation mapping (Grad-CAM) visualization achieved high accuracy in brain cancer detection and localization [141]. Grad-CAM provided heatmaps overlaying the most relevant tumor regions identified by the model, offering clinicians a direct view of the areas driving predictions.

The importance of XAI lies in its ability to visualize model features, improve interpretability, and enable human–machine interactions that are crucial for clinical adoption [161]. Recent developments, such as NeuroXAI, a framework implementing multiple XAI methods for MRI analysis of brain tumors, demonstrate the potential of XAI to enhance transparency and reliability in neuroimaging, facilitating the adoption of AI models in clinical practice for cancer diagnosis and treatment planning [162].

Another study explored the adaptive aquila optimizer with XAI for effective colorectal and osteosarcoma cancer classification, combining faster SqueezeNet for feature extraction, adaptive optimization for tuning, ensemble DL classifiers for diagnosis, and LIME for interpretability [163].

## 9. Discussion

AI has the potential to transform medicine by enabling the analysis of large quantities of patient data to provide faster, more accurate diagnoses and to reduce the need for invasive procedures. AI can also help healthcare providers to optimize clinical workflows by automating repetitive tasks and reducing the administrative burden on clinicians.

A variety of machine learning techniques has been applied to address key challenges in pediatric neuroimaging for cancer. Table 2 provides an overview of these key issues, their descriptions, and the ML solutions currently being explored. This structured summary highlights limitations in dataset availability, tumor detection, scan time reduction, and treatment outcome prediction, emphasizing the role of AI-driven solutions such as deep learning, radiomics, and federated learning.

### 9.1. Data-Related Issues

A current limitation on using AI for neuroimaging applications in pediatric cancer is the lack of large datasets, as AI models require a substantial amount of data to learn. One way to overcome this limitation is to encourage collaboration and data sharing among laboratories across universities, research organizations, hospitals, and other healthcare institutions. This can be achieved by developing standardized protocols for data collection, processing, and analysis, as well as by establishing data sharing agreements that ensure data privacy and ethical use of data. An alternative approach to address the problem of limited datasets is transfer learning, in which pre-trained models from other domains or populations can be fine-tuned for use with smaller pediatric datasets.

Another limitation of pediatric neuroimaging is the lack of standardization in imaging protocols and quality control measures across different research laboratories. If non-standardized datasets are combined in an AI model, such differences can decrease the accuracy and reliability of the model. To streamline image processing and analysis, researchers and clinicians can leverage tools such as SPM [164] for statistical analysis, FSL [165] for multi-modal imaging, and FreeSurfer [166] for automated cortical and subcortical segmentation. These tools address common challenges like motion artifacts and variability in pediatric brain development.

### 9.2. Technical Limitations

There are several knowledge gaps concerning AI for neuroimaging applications in pediatric cancer. First, although AI models have shown promising results in diagnosing and predicting outcomes of some pediatric neurological diseases, such as brain tumors, there has been only limited research on the generalizability of these models across different patient populations.

Second, there is a need for more research on the interpretability of AI models in pediatric neuroimaging. Interpretability of AI models is critical, especially in healthcare applications such as pediatric neuroimaging. Although AI models may provide accurate predictions, the black-box nature of some AI models can be a concern, as it may limit the ability of the user to understand how the model arrives at its decisions or predictions, thereby potentially limiting their clinical utility. Developing methods to explain the logic behind AI model predictions would help to improve their transparency, trustworthiness, and scientific contribution [145].

Third, there is a need for more research on integrating AI models into clinical practice. The development of AI models is only the first step in their implementation into clinical workflows, and more research is needed to determine how these models can be integrated into clinical decision making and how they can have an impact on patient outcomes.

Explainable AI is paramount in pediatric neuroimaging for several reasons. In clinical applications, understanding the decision-making process of AI models is crucial for gaining trust among healthcare professionals and facilitating seamless integration into diagnostic workflows. Moreover, in cancer, where decisions can have profound implications, explainability ensures that clinicians and researchers comprehend how AI arrives at its predictions or classifications. The application of AI in pediatric neuroimaging demands a high level of interpretability. This transparency fosters a collaborative environment between AI tools and healthcare practitioners. As we navigate the future of AI in pediatric neuroimaging, emphasis should be placed on furthering the development of explainable AI techniques. This not only aligns with the growing demand for transparency in AI applications but also positions pediatric catastrophic disease research at the forefront of responsible AI implementation.

### 9.3. Ethical Considerations

Ethical concerns such as data privacy, bias, and transparency need to be addressed in the development and implementation of AI models for pediatric neuroimaging. Ensuring that these models are developed and validated in an ethical and responsible manner is essential to avoid potential harm to patients and maintain public trust in AI applications. Practical solutions to address privacy concerns include frameworks for multi-institutional data sharing and the potential of federated learning, which allows for collaborative model training without transferring raw data, and blockchain [167]. Establishing guidelines and ethical frameworks for the responsible use of AI in patient care and research will be crucial as AI becomes more prominent in neuroimaging.

### 9.4. Future Directions

AI can facilitate the integration and analysis of multimodal neuroimaging data from various sources and research centers [168,169]. Collaborative efforts with AI-based tools can accelerate discoveries and improve data sharing within the scientific community. AI can help identify novel biomarkers associated with various neurological and psychiatric conditions by analyzing large datasets of functional imaging data. These biomarkers could aid in early diagnosis and personalized treatment strategies. Advanced AI algorithms can predict the progression of neurological disorders and their response to specific neuromodulation treatments. This predictive capability can help clinicians to make more informed decisions about patient care and long-term management.

The use of AI for pediatric neuroimaging is a rapidly growing field. Some of the key areas of focus for future developments in this field include large-scale data collection, multimodal integration of neuroimaging modalities (such as MRI, fMRI, resting-state fMRI, MRS, DTI, fNIRS, EEG, and MEG), early detection of pathologies, real-time neuroimage processing and analysis, personalized treatment planning, prediction of treatment outcomes, longitudinal neuroimaging analysis, and the theoretical explanation of pathological phenomena.

When studying brains that were previously affected by solid tumors, the approaches used should consider (1) mass effect, with reference to displacement and compression indices; (2) edema, with reference to morphometry, density, and composition; and (3) maps of tissue damage, in terms of volume, morphometry, density, and structural/functional connectivity. Density can be extracted from T1 MRI images, whereas composition is derived from PET scans. Structural connectivity analysis may consider diffusion imaging features extracted as voxel-based measures, such as the apparent diffusion coefficient, fractional anisotropy, mean diffusivity, radial diffusivity, and axial diffusivity, and tensor-based measures, such as tensor connectivity maps.

AI can further refine and enhance the analysis of functional imaging data, such as fMRI. Advanced AI algorithms can extract more precise information from complex brain activity patterns and lead to better insights into brain function and connectivity. For pediatric neuroimaging to truly benefit patient care, AI applications must extend beyond research settings to real-time clinical relevance. AI-driven tools can expedite data analysis, aiding in early detection, diagnosis, and treatment planning. The development of closed-loop systems, guided by real-time functional imaging data, showcases the potential of AI to dynamically adapt neuromodulation strategies, ensuring personalized and efficient interventions for pediatric patients.

AI can enable closed-loop systems in which real-time functional imaging data is used to adapt and optimize neuromodulation in real time. This dynamic approach can ensure that stimulation parameters are continuously adjusted to suit the changing state of the brain, making treatments more efficient and effective. AI can be used in designing and optimizing neuromodulation devices, such as DBS systems. AI-driven simulations can assist in developing more precise and targeted stimulation paradigms. Moreover, AI can assist in identifying the most effective neuromodulation techniques for individual patients, based on their unique brain activity patterns. By considering patient-specific features, such as neural network connectivity, AI can optimize treatment parameters for better outcomes.

## 10. Conclusions

Although AI has seen significant advances in its application to neuroimaging in adult populations, its implementation in children with cancer has been limited by several factors, including the scarcity of available datasets and the unique challenges of applying adult-focused AI methods to pediatric populations. The limited number of pediatric neuroimaging datasets poses a significant challenge in training AI models specifically for children with cancer. AI algorithms rely heavily on large, diverse datasets to learn patterns and make accurate predictions. Additionally, the neuroimaging characteristics of children are different from those of adults because of ongoing brain development, size differences, and diverse neurological conditions that may manifest differently in pediatric patients. Therefore, direct translation of AI approaches from adult neuroimaging to children might not be feasible without appropriate adjustments and validation. Conversely, AI models trained on a restricted pool of pediatric data might be less effective than those trained on larger and more comprehensive datasets derived from adults.

To address these challenges and to foster the development of AI in pediatric neuroimaging, there is a need to build larger and more diverse pediatric neuroimaging databases. Collaboration among institutions and data sharing initiatives are essential to ensure the responsible and effective use of AI in pediatric populations. As collaborations and data sharing initiatives begin to standardize pediatric data collection, the development of AI approaches tailored specifically to pediatric applications will increase significantly.

## Figures and Tables

**Figure 1 cancers-17-00622-f001:**
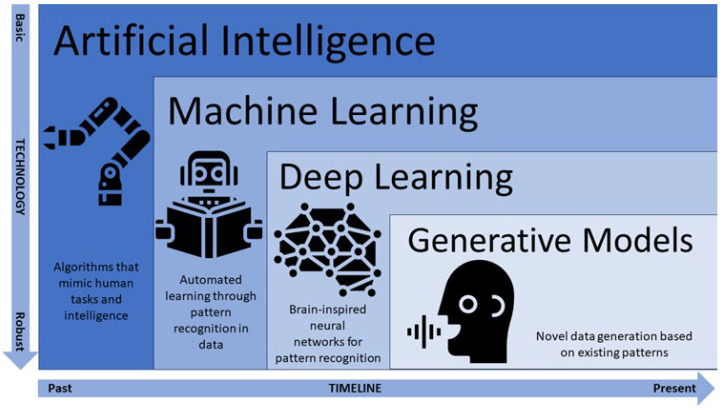
Chronological development of sub-categories and approaches in artificial intelligence.

**Figure 2 cancers-17-00622-f002:**
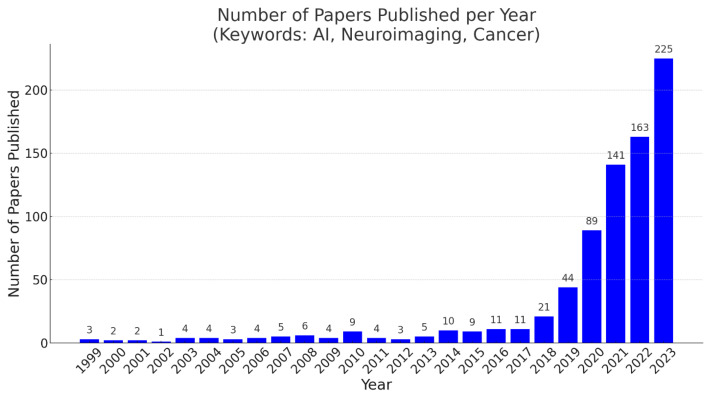
Number of papers published per year containing the keywords “artificial intelligence”, “neuroimaging”, and “cancer”, according to the ScienceDirect search tool (www.sciencedirect.com/search (accessed on 31 October 2024).

**Figure 3 cancers-17-00622-f003:**
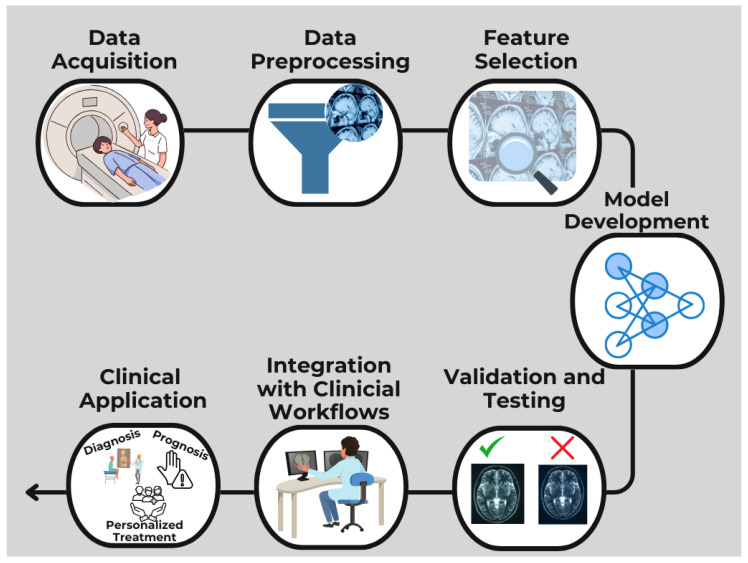
Workflow for artificial intelligence based pediatric neuroimaging.

**Table 1 cancers-17-00622-t001:** Summary of key studies evaluating AI-based tumor segmentation models in pediatric neuroimaging, including dataset size details and performance metrics.

Study	Dataset Size	Performance
[102]	Training: 184 samples/testing: 100 samples	Dice coefficient: 0.877
[103]	Training: 261 samples/testing: 112 samples	Dice coefficient: 0.657–0.967
[104]	Training: 261 samples/testing: 30 samples	Dice coefficient: 0.642
[105]	Training: 364 samples/testing: 125 samples	Dice coefficient: 0.98

**Table 2 cancers-17-00622-t002:** Key issues and respective machine learning solutions in AI-based pediatric neuroimaging for cancer.

Key Issue	Description	Machine Learning Solutions
Limited pediatric data	Scarcity of large, high-quality datasets impedes AI model training.	Transfer learning, federated learning, GANs for data augmentation
Motion artifacts in young patients	Children struggle to stay still during scans, leading to degraded image quality.	AI-based motion correction, CNNs for image denoising
Prolonged scan times	Long MRI scans increase discomfort and require sedation.	Compressed sensing, AI-accelerated MRI reconstruction (e.g., deep CNNs)
Radiation and contrast agent exposure	Pediatric patients are more sensitive to radiation and contrast dyes.	Low-dose imaging with deep learning, contrast-free MRI enhancement with CNNs
Tumor detection and segmentation	Early and precise tumor localization is essential for treatment planning.	CNNs (3D-CNN, ResNet, U-Net, nnU-Net), GANs for data augmentation
Tumor classification and molecular subtyping	Differentiating between tumor types and predicting molecular markers.	Radiomics radiogenomics, CNNs, SVMs, deep learning classifiers
Functional neuroimaging and cognitive deficits	Pediatric brain tumors affect cognitive function, requiring fMRI and other functional imaging.	AI-based functional MRI analysis, machine learning for cognitive pattern recognition (e.g., SVM, CNNs)
Explainability and interpretability of AI	Clinicians need interpretable AI models for trust and adoption.	Explainable AI (XAI), SHAP, LIME, Grad-CAM
Personalized treatment planning	Individualized therapy selection based on imaging and genetic markers.	Multi-modal deep learning (integrating MRI, PET, clinical data)
Predicting treatment outcomes and survival	AI models need to distinguish between real tumor progression and pseudoprogression.	Radiomic feature analysis, deep learning prognostic models, survival prediction frameworks

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
