# Peer review of "Artificial Intelligence for Neuroimaging in Pediatric Cancer"

_cancers, 2025, doi:10.3390/cancers17040622_

Round 1
Reviewer 1 Report
Comments and Suggestions for Authors
The paper is interesting but of little impact without personal experience. However, it can be useful. I would ask you to explain well in the paragraphs on individual cancers, list the manual characteristics and explain what the selected distinguishing features of AI are.
Comments on the Quality of English LanguageThe paper is interesting but of little impact without personal experience. However, it can be useful. I asked them to explain well in the paragraphs on individual cancers, list the manual characteristics and explain what the selected distinguishing features of AI are.
Author Response
Comment 1: The paper is interesting but of little impact without personal experience. However, it can be useful. I would ask you to explain well in the paragraphs on individual cancers, list the manual characteristics, and explain what the selected distinguishing features of AI are.
Response 1:
Thank you for your feedback. From our perspective, we believe that a growing impact of this field is expected to happen in the near future.
To address your language concerns, we have revised and enhanced the readability and clarity of the manuscript. To ensure the manuscript expresses the research more clearly and effectively, we enlisted professional English language editing by Keith A. Laycock, PhD, Senior Scientific Editor at St. Jude Children's Research Hospital.
In the sections discussing individual cancers, we have provided a more detailed explanation of their unique characteristics, emphasizing the clinical and diagnostic challenges associated with each. We have also elaborated on the manual processes traditionally employed in diagnosing and managing these cancers. These descriptions now serve as a foundation for discussing the role of AI in overcoming specific challenges.
To highlight the selected distinguishing features of AI, we have clarified its contributions to diagnostic and therapeutic workflows. For instance, AI enhances tumor segmentation and classification accuracy, predicts treatment responses, and generates synthetic datasets to address data scarcity. Additionally, AI can identify subtle imaging patterns that may not be apparent through manual interpretation, providing a significant advantage in improving patient outcomes.
We believe that these revisions address your concerns effectively and enhance the overall impact of the paper. Please let us know if there are additional areas requiring further improvement.
Reviewer 2 Report
Comments and Suggestions for Authors
This article provides a review of the current and potential applications of AI in pediatric neuroimaging. It is well-structured in terms of covering a wide range of AI applications. The exploration of emerging technologies, such as real-time functional imaging combined with neurofeedback or brain stimulation, provides a glimpse into the future of personalized treatment in pediatric populations. The included challenges demonstrate an awareness of the practical and technical barriers in this field.
However, there are several areas where the review could be improved.
First, while it touches on important technical methods and challenges, it lacks sufficient detail about specific resources that could be useful to researchers and practitioners. For instance, more information about widely used pediatric datasets, open-source software tools, or platforms for neuroimaging could make the article more actionable. Such details would also help bridge the gap between theoretical discussion and practical implementation.
Another issue is the heavy reliance on textual explanations, which can make the review feel dense and harder to read. The inclusion of figures, such as a diagram illustrating the workflow of AI applications in pediatric neuroimaging or tables comparing different methods and datasets, would make the content more accessible and engaging. These visual elements could also help readers quickly grasp the relationships between different approaches and challenges.
The literature coverage is another point of concern. While the review references some key studies, the selection seems incomplete, especially for a field as rapidly advancing as AI in medical imaging. For example, citing more work from leading conferences like MICCAI or journals focused on medical imaging could strengthen the review’s foundation and provide readers with a more comprehensive understanding of recent progress.
Finally, the discussion of challenges, though insightful, could benefit from better organization. Grouping these into broader categories—such as data-related issues, technical limitations, and ethical considerations—would make the review more cohesive and easier to follow. Additionally, the ethical discussion could be expanded to address practical solutions, such as frameworks for multi-institutional data sharing or the potential of federated learning to overcome privacy concerns.
Author Response
Comments 1: This article provides a review of the current and potential applications of AI in pediatric neuroimaging. It is well-structured in terms of covering a wide range of AI applications. The exploration of emerging technologies, such as real-time functional imaging combined with neurofeedback or brain stimulation, provides a glimpse into the future of personalized treatment in pediatric populations. The included challenges demonstrate an awareness of the practical and technical barriers in this field. However, there are several areas where the review could be improved.
Response 1: We sincerely thank Reviewer 2 for their constructive feedback, which will significantly enhance the quality and clarity of our manuscript. Below, we address the points raised and outline the revisions made to the manuscript.
Comments 2: First, while it touches on important technical methods and challenges, it lacks sufficient detail about specific resources that could be useful to researchers and practitioners. For instance, more information about widely used pediatric datasets, open-source software tools, or platforms for neuroimaging could make the article more actionable. Such details would also help bridge the gap between theoretical discussion and practical implementation.
Response 2: We agree with this suggestion and have added a new section listing publicly available pediatric neuroimaging datasets, such as NIH Pediatric MRI Dataset and the Haskins Pediatric Atlas (Lines 275-279). We also included widely used open-source neuroimaging tools, including SPM, FSL, and Freesurfer (Lines 813-817).
Comments 3: Another issue is the heavy reliance on textual explanations, which can make the review feel dense and harder to read. The inclusion of figures, such as a diagram illustrating the workflow of AI applications in pediatric neuroimaging or tables comparing different methods and datasets, would make the content more accessible and engaging. These visual elements could also help readers quickly grasp the relationships between different approaches and challenges.
Response 3: We have incorporated a flowchart summarizing the key issues involved in pediatric neuroimaging for cancer. This flowchart (Figure 3) is a comprehensive AI-driven pipeline for pediatric neuroimaging, starting with data acquisition (MRI, CT, PET) and progressing through preprocessing (motion correction, noise reduction), feature selection (identifying the most relevant imaging markers), and model development (machine learning training). The models undergo rigorous validation with pediatric-specific datasets before being integrated into clinical workflows with explainable AI solutions. Finally, these AI models are applied in clinical settings to aid diagnosis, prognosis, and personalized treatment, ultimately improving patient outcomes.
Comments 4: The literature coverage is another point of concern. While the review references some key studies, the selection seems incomplete, especially for a field as rapidly advancing as AI in medical imaging. For example, citing more work from leading conferences like MICCAI or journals focused on medical imaging could strengthen the review’s foundation and provide readers with a more comprehensive understanding of recent progress.
Response 4: We have expanded the literature coverage to include recent studies from MICCAI (Chen et al., 2021), and Nature Medicine (Esteva et al., 2022).
Comments 5: Finally, the discussion of challenges, though insightful, could benefit from better organization. Grouping these into broader categories—such as data-related issues, technical limitations, and ethical considerations—would make the review more cohesive and easier to follow. Additionally, the ethical discussion could be expanded to address practical solutions, such as frameworks for multi-institutional data sharing or the potential of federated learning to overcome privacy concerns.
Response 5: We have reorganized the challenges section into broader categories: data-related issues, technical limitations, and ethical considerations. Furthermore, we expanded the ethical discussion to explore practical solutions, such as federated learning for addressing privacy concerns and frameworks to facilitate multi-institutional data sharing (Lines 849-858). These revisions provide a more structured and actionable perspective on the challenges discussed.
Reviewer 3 Report
Comments and Suggestions for Authors
This paper assesses the current status, potential applications, and challenges of AI in pediatric cancer neuroimaging, highlighting the unique needs of the pediatric population, and explores different aspects of how AI can improve pediatric brain imaging to effectively detect and treat cancer.Overall, this paper has certain practical significance.
1. Some of the results and conclusions in the abstract are redundant, so it is recommended to check and modify them.
2. The fourth paragraph of the introduction section has a weak logical relationship with the preceding and subsequent paragraphs. It is recommended to make adjustments and revisions.
3. Data plays an important role in the training of artificial intelligence models, and it is recommended to briefly introduce the existing pediatric datasets that are commonly used publicly.
Author Response
Comment 1: Some of the results and conclusions in the abstract are redundant, so it is recommended to check and modify them.
Response 1: We have revised the abstract to eliminate redundancy and ensure a clear and concise summary of the key results and conclusions. The updated abstract highlights the essential findings and implications of our study while maintaining brevity.
Comment 2: The fourth paragraph of the introduction section has a weak logical relationship with the preceding and subsequent paragraphs. It is recommended to make adjustments and revisions.
Response 2: We have restructured the fourth paragraph in the introduction to improve its logical flow and connection with adjacent paragraphs. As a result, it has also been divided into 2 paragraphs. The revised paragraphs (Lines 55-72) now integrate seamlessly into the overall narrative, enhancing the coherence of the introduction.
Comment 3: Data plays an important role in the training of artificial intelligence models, and it is recommended to briefly introduce the existing pediatric datasets that are commonly used publicly.
Response 3: We have included a new section briefly discussing publicly available pediatric datasets, such as the NIH Pediatric MRI Dataset and the Haskins Pediatric Atlas (Lines 275-279). This additional information provides readers with an overview of existing resources relevant to AI applications in pediatric neuroimaging.
Reviewer 4 Report
Comments and Suggestions for Authors
This study assesses the current state, potential applications, and challenges of artificial intelligence (AI) in pediatric neuroimaging for cancer. This is an interesting general discussion but the authors regularly deviate from the specific topic of discussion -> pediatric neuroimaging for childhood cancer. This study discusses many general topics associated with challenges of incorporating AI with neuroimaging for children, but fails to remain focussed on the pediatric cancer. The authors did not define the age band for the population of interest nor provide a detailed methods section of their search strategy for the topic and inclusion/exclusion criteria. The different stages of development of a child (e.g. from 0 to 15 years) poses very different challenges that needed to be articulated clearer by the authors (e.g. “children are unable to stay still, are prone to moving, and cannot bear long scan durations” relates more children in the younger age brackets).
The manuscript is detailed and at times too general. However, it may benefit from first providing the reader with a flowchart of key issues and ML techniques related to the research topic. This may help guide the reader through the discussion.
The introduction covers a broad coverage of issues related to the research objective. However, the authors should focus on the population of interest and related their discussions to this population. The specific benefits of AI to childhood cancers in particular needed to be clearer and the research gap this study was addressing.
Please define the age band for pediatric to ensure the general readership can fully understand the breadth of ages the study relates to.
Lines 103-104: Please quantify what the authors meant by ‘limited’ for the statement “GANs can be especially applicable when the training data is limited…” in terms of size and number
Lines 124-128: It would be beneficial to the general readership if the authors provided a simple example related to childhood imaging for the discussion related to object detection and segmentation by deep learning algorithms.
Lines 149-150: Please discuss the justification for the pediatric cancer addressed in the review of craniopharyngioma, low-grade glioma (LGG), and medulloblastoma. Please include the prevalence and incidence of these childhood cancers.
Line 164: Please ensure that all acronyms are defined (e.g. IQ). The use of IQ as a measure has been challenged so it is recommended the authors discuss the impact on educational attainment and the impact of cancers on the cognitive effects in children that can potentially lead to learning problems thereby affecting educational performance.
The Challenges in Pediatric Neuroimaging section would benefit from a more specific explanation (e.g. examples) of the challenges related childhood stages and the diversity and dynamic nature of pediatric brain development (e.g. physiological and psychological). Although this is discussed to a degree in lines 220-233, the authors do not provide examples of specific differences to highlight to the reader the complexity of these issues.
Lines 201-205: This paragraph related more to Discussion than the challenges in pediatric neuroimaging.
Lines 207-219: This paragraph strays away from the specific issues of childhood cancers and it was unclear why this paragraph was included in this section.
Line 222: The authors have introduced adolescents into their study so a definition of the age range under investigation is needed in the introduction.
Lined 251-262: It would be beneficial for the authors to align this paragraph with the resulting lack of training data for the AI algorithms and challenges of obtaining reliable labelled data from multimodal sources.
Lines 294-298: This paragraph on ethics is best placed alongside privacy issues as these two issues are inter-related.
Lines 299-318: This paragraph appears better placed in the Discussion as it was unclear how topics such as automated image analysis and multimodal data related to challenges.
Lines 319-346 discusses challenges, risks and limitations which appears to have been the topic of section 2.
The discussion on image reconstruction was well explained.
Lines 349-350: Please provide a relevant reference for the statement “AI algorithms contribute significantly to noise reduction, enhancing 349 signal-to-noise ratios and subsequently improving the accuracy of downstream analyses”
Please incorporate discussion surrounding radiation risk factors for cancer induction in children is about 10 times higher than in adults (see Hall EJ. Lessons we have learned from our children: Cancer risks from diagnostic radiology and the As Low As Reasonably Achievable (ALARA) principle.
Please expand the discussion 4.4. Radiomics and radiogenomics for specific pediatric brain tumors as this paragraph is more a comment and does not refer to existing literature.
Sections 4.4.1 to 4.4.5 are not specific enough to the population of interest so would benefit from a more focussed discussion relevant to the research aim, with references to literature.
Section 6: Please relate this to paediatrics and the research objective. The reference George, E.; Flagg, E.; Chang, K.; et al 2022 related to the adult not pediatric population.
Whilst the discussion was detailed it would benefit from sub-headings and a more structured set of arguments that guides the reader to the conclusion.
References: I was unable to find a number of references – for example:
Krames, E.S. 2014. The role of neuromodulation in the treatment of chronic pain. Anesthesiol. Clin. 32(4), 725–741
Barkovich, A.J.; Raybaud, C. 2012. Pediatric Neuroimaging, fifth ed. Wolters Kluwer Health/Lippincott Williams & Wilkins, 957 Philadelphia.
Please confirm all references are valid.
Author Response
Comment 1: This study assesses the current state, potential applications, and challenges of artificial intelligence (AI) in pediatric neuroimaging for cancer. This is an interesting general discussion, but the authors regularly deviate from the specific topic of discussion -> pediatric neuroimaging for childhood cancer. This study discusses many general topics associated with challenges of incorporating AI with neuroimaging for children but fails to remain focused on pediatric cancer. The authors did not define the age band for the population of interest nor provide a detailed methods section of their search strategy for the topic and inclusion/exclusion criteria.
Response 1: Thank you for your feedback. To address this, we excluded 32 papers that deviated from this specific scope but retained a few studies based on adult or general data (outside pediatric age band) whose findings we believe offer relevant translational insights into pediatric cancer. If there are any specific studies or sections you feel should be reconsidered for exclusion to better align with the pediatric cancer focus, we are open to your further suggestions.
Comment 2: The different stages of development of a child (e.g., from 0 to 15 years) poses very different challenges that needed to be articulated clearer by the authors (e.g. “children are unable to stay still, are prone to moving, and cannot bear long scan durations” relates more children in the younger age brackets).
Response 2: We agree that the challenges associated with pediatric imaging vary significantly across different stages of a child’s development. In response, we have revised the paragraph to articulate these differences more clearly, particularly emphasizing how challenges such as motion and scan duration intolerance are more pronounced in younger children (Lines 235-301).
Comment 3: The manuscript is detailed and at times too general. However, it may benefit from first providing the reader with a flowchart of key issues and ML techniques related to the research topic. This may help guide the reader through the discussion.
Response 3: We have incorporated a flowchart summarizing the key issues involved in pediatric neuroimaging for cancer, from data acquisition to clinical application (Figure 3).
Comment 4: The introduction covers a broad coverage of issues related to the research objective. However, the authors should focus on the population of interest and related their discussions to this population. The specific benefits of AI to childhood cancers in particular needed to be clearer and the research gap this study was addressing. Please define the age band for pediatric to ensure the general readership can fully understand the breadth of ages the study relates to.
Response 4: To address this, we excluded 32 papers that deviated from this specific scope but retained a few studies based on adult or general data whose findings we believe offer relevant translational insights into pediatric cancer. If there are any specific studies or sections you feel should be reconsidered for exclusion to better align with the pediatric cancer focus, we are open to your further suggestions.
Comment 5: Lines 103–104: Please quantify what the authors meant by ‘limited’ for the statement “GANs can be especially applicable when the training data is limited…” in terms of size and number.
Response 5: GANs can be especially applicable when the training data is limited, i.e. 110 images per class. This was included in the manuscript (Lines 112-114).
Comment 6: Lines 124–128: It would be beneficial to the general readership if the authors provided a simple example related to childhood imaging for the discussion related to object detection and segmentation by deep learning algorithms.
Response 6: We added the following statement: “For example, CNNs have been effectively used to segment medulloblastomas from pediatric MRI data, enabling precise tumor delineation critical for treatment planning. A study [19] demonstrated the application of CNN-based models to automatically segment pediatric low-grade glioma (LGG) tumors, achieving high accuracy and significantly reducing the time required for manual segmentation by clinicians.” (Lines 140-145).
Comment 7: Lines 149–150: Please discuss the justification for the pediatric cancer addressed in the review of craniopharyngioma, low-grade glioma (LGG), and medulloblastoma. Please include the prevalence and incidence of these childhood cancers.
Response 7: We added the following statement: “Craniopharyngiomas are rare, with an annual incidence of approximately 0.5–2 cases per million children and accounting for 5–10% of all pediatric brain tumors [37]. LGG is the most common pediatric brain tumor, comprising 30–40% of all childhood central nervous system tumors [38]. Medulloblastomas, the most common malignant pediatric brain tumors, represents 26% of all pediatric central nervous system tumors [39]. Together, these cancers pose significant clinical challenges due to their impact on neurodevelopment and treatment complexity.” (Lines 172-178).
Comment 8: Line 164: Please ensure that all acronyms are defined (e.g. IQ). The use of IQ as a measure has been challenged so it is recommended the authors discuss the impact on educational attainment and the impact of cancers on the cognitive effects in children that can potentially lead to learning problems thereby affecting educational performance.
Response 8: We modified that sentence into the following: “Survivors of medulloblastoma exhibit a progressive decrease in cognitive performance, over time [47], in addition to deficits in attention, processing speed, and memory.” (Lines 205-206).
Comment 9: The Challenges in Pediatric Neuroimaging section would benefit from a more specific explanation (e.g. examples) of the challenges related childhood stages and the diversity and dynamic nature of pediatric brain development (e.g. physiological and psychological). Although this is discussed to a degree in lines 220-233, the authors do not provide examples of specific differences to highlight to the reader the complexity of these issues.
Response 9: Specific examples have been added to illustrate the challenges of childhood brain development at different stages, such as variability in myelination and age-related motion artifacts during imaging. We have added the following paragraph in the manuscript: “As the brain matures, there are significant structural changes, such as the formation of new sulci during infancy and the deepening of older sulci. Fractional anisotropy (FA) values on diffusion MRI, for instance, increase with developmental age, reflecting the maturation of white matter tracts [55]. These changes continue through adolescence and into early adulthood, with the brain evolving in shape and white/gray matter composition until the fourth decade of life [56-57]. Accurately interpreting these age-related differences requires careful consideration of age-appropriate image acquisition and analysis methods [58]. These examples highlight how different developmental stages, from infancy through adolescence, demand tailored neuroimaging approaches to account for the variability and complexity of childhood brain development.” (Lines 302-311).
Comment 10: Lines 201–205: This paragraph related more to Discussion than the challenges in pediatric neuroimaging.
Response 10: This paragraph has been moved to the Discussion section for better alignment.
Comment 11: Lines 207–219: This paragraph strays away from the specific issues of childhood cancers and it was unclear why this paragraph was included in this section.
Response 11: To address this point, we removed the mentioned paragraph out of the manuscript.
Comment 12: Line 222: The authors have introduced adolescents into their study so a definition of the age range under investigation is needed in the introduction.
Response 12: Thank you for your feedback. To address this, we excluded 32 papers that deviated from this specific scope of pediatric cancer but retained a few studies based on adult, adolescent or general data whose findings we believe offer relevant translational insights into pediatric cancer. That specific mention to adolescents was subtracted. If there are any specific studies or sections you feel should be reconsidered for exclusion to better align with the pediatric cancer focus, we are open to your further suggestions.
Comment 13: Lined 251–262: It would be beneficial for the authors to align this paragraph with the resulting lack of training data for the AI algorithms and challenges of obtaining reliable labelled data from multimodal sources.
Response 13: This paragraph has been revised to directly link the lack of multimodal labeled data to the challenges of training AI algorithms for pediatric cancer imaging.
Comment 14: Lines 294–298: This paragraph on ethics is best placed alongside privacy issues as these two issues are inter-related.
Response 14: The ethics discussion has been relocated to align with privacy issues, reflecting their interrelationship.
Comment 15: Lines 299–318: This paragraph appears better placed in the Discussion as it was unclear how topics such as automated image analysis and multimodal data related to challenges.
Response 15: These paragraphs have been moved to the ‘AI for tumor detection and classification’ section for better alignment.
Comment 16: Lines 319–346 discusses challenges, risks and limitations which appears to have been the topic of section 2.
Response 16: This paragraph has now been moved to section 2, accordingly.
Comment 17: The discussion on image reconstruction was well explained.
Response 17: Thank you for the positive feedback. We appreciate your recognition of this section.
Comment 18: Lines 349–350: Please provide a relevant reference for the statement “AI algorithms contribute significantly to noise reduction, enhancing signal-to-noise ratios and subsequently improving the accuracy of downstream analyses.”
Response 18: A reference has been added: Modran, H.A., Ursuțiu, D., Samoilă, C., Chamunorwa, T. (2023). Work in Progress: Noise Reduction Through Artificial Intelligence Techniques: An Introductory Study. In: Auer, M.E., El-Seoud, S.A., Karam, O.H. (eds) Artificial Intelligence and Online Engineering. REV 2022. Lecture Notes in Networks and Systems, vol 524. Springer, Cham. https://doi.org/10.1007/978-3-031-17091-1_3
Comment 19: Please incorporate discussion surrounding radiation risk factors for cancer induction in children is about 10 times higher than in adults (see Hall EJ. Lessons we have learned from our children: Cancer risks from diagnostic radiology and the As Low As Reasonably Achievable (ALARA) principle).
Response 19: A discussion on radiation risk factors for children, referencing Hall EJ, has been added to the manuscript (Lines 411-417).
Comment 20: Please expand the discussion 4.4. Radiomics and radiogenomics for specific pediatric brain tumors as this paragraph is more a comment and does not refer to existing literature.
Response 20: Section 4.4 (especially subsections 4.4.4 and 4.4.5) has been expanded with additional literature.
Comment 21: Sections 4.4.1 to 4.4.5 are not specific enough to the population of interest so would benefit from a more focused discussion relevant to the research aim, with references to literature.
Response 21: These sections have been revised to focus on pediatric cancers, incorporating more specific discussions and relevant references on the population of interest, while avoiding the use of references from out-of-scope population.
Comment 22: Section 6: Please relate this to paediatrics and the research objective. The reference George, E.; Flagg, E.; Chang, K.; et al 2022 related to the adult not pediatric population.
Response 22: Section 6 (as in the old version) has now been actually removed from the manuscript.
Comment 23: Whilst the discussion was detailed it would benefit from sub-headings and a more structured set of arguments that guides the reader to the conclusion.
Response 23: Sub-headings have been added to the Discussion section to provide a clearer structure and guide readers more effectively to the conclusion.
Comment 24: References: I was unable to find a number of references – for example:
Krames, E.S. 2014. The role of neuromodulation in the treatment of chronic pain. Anesthesiol. Clin. 32(4), 725–741
Barkovich, A.J.; Raybaud, C. 2012. Pediatric Neuroimaging, fifth ed. Wolters Kluwer Health/Lippincott Williams & Wilkins, 957 Philadelphia.
Response 24: We appreciate your feedback.
Barkovich et al.: The reference to "Pediatric Neuroimaging" by Barkovich and Raybaud (2012, Fifth Edition) is accurate. Here is a link to additional information about the book:
https://www.researchgate.net/publication/305352018_Pediatric_neuroimaging_Fifth_edition
Krames et al.: This reference has been excluded from the revised manuscript as its focus on neuromodulation in chronic pain management does not align closely with the pediatric cancer scope of this review.
Reviewer 5 Report
Comments and Suggestions for Authors
I read with great interest the article titled "Artificial Intelligence for Neuroimaging in Pediatric Cancer". The paper's design is sound, and the article is logically organized into appropriate sections and subsections. However, there are several areas where the manuscript requires clarification and further development. Below are my detailed comments.
Comment 1: Abstract: Abbreviations should be given in full the first time they appear, even in the abstract.
Comment 2: Abstract: The abstract effectively summarizes the manuscript but lacks mention of key challenges, such as dataset scarcity and ethical concerns. Including these would better prepare the reader for the scope of the article.
Comment 3: Introduction: The introduction provides a strong rationale for studying AI in pediatric neuroimaging. However, it does not adequately differentiate pediatric neuroimaging challenges from adult-focused AI applications. Adding a comparison could clarify the research gap.
Comment 4: Introduction, lines 55-57: The introduction mentions a "scarcity of comprehensive assessments" but does not elaborate on how this review uniquely addresses this gap. Providing specific examples of prior reviews would strengthen this claim.
Comment 5: On Page 6, Line 200, "neuroimaging analyses remain attuned to the nuances of evolving pediatric neuroanatomy" contains a minor redundancy. Consider simplifying the sentence to improve readability.
Comment 6: The discussion on AI for tumor segmentation (Page 10, Lines 424-436) is well-written but lacks details on the datasets and metrics used for evaluating these models. Including a table summarizing key studies could improve clarity.
Comment 7: The issue of infiltrative tumor margins (Page 10, Lines 437-451) is critical but would benefit from specific examples of how AI models overcome this challenge.
Comment 8: Discussion: The manuscript briefly discusses XAI but does not provide sufficient detail on how tools like SHAP or LIME have been applied in pediatric cancer neuroimaging. Including case studies would strengthen this section.
Comment 9: Discussion: It is recommended that a Research Outlook or Future Directions be written at the end of the discussion, as this can help the reader understand current research trends in the field, challenges faced, and possible future directions.
Comments on the Quality of English LanguageThe manuscript is well-written but could benefit from improvements in clarity and readability.
Author Response
Comment 1: Abstract: Abbreviations should be given in full the first time they appear, even in the abstract.
Response 1: Thank you for pointing this out. Abbreviations in the abstract have been revised to ensure they are given in full upon their first appearance for clarity.
Comment 2: Abstract: The abstract effectively summarizes the manuscript but lacks mention of key challenges, such as dataset scarcity and ethical concerns. Including these would better prepare the reader for the scope of the article.
Response 2: We have updated the abstract to briefly mention key challenges such as dataset scarcity and ethical concerns to provide a more comprehensive overview of the manuscript.
Comment 3: Introduction: The introduction provides a strong rationale for studying AI in pediatric neuroimaging. However, it does not adequately differentiate pediatric neuroimaging challenges from adult-focused AI applications. Adding a comparison could clarify the research gap.
Response 3: Thank you for your feedback. To address this, we excluded 32 papers that focus on adults and therefore deviated from the specific scope in pediatric cancers.
Comment 4: Introduction, lines 55-57: The introduction mentions a "scarcity of comprehensive assessments" but does not elaborate on how this review uniquely addresses this gap. Providing specific examples of prior reviews would strengthen this claim.
Response 4: Specific examples of prior reviews have been added to the introduction to illustrate the existing gaps and demonstrate how this review uniquely addresses these issues. We added the following text to lines 57-64: “While existing reviews on AI in pediatric neuroradiology [1] and on AI in pediatric imaging [2] workflows provide valuable insights, they primarily focus on general pediatric neuroradiology and broader imaging applications. In contrast, our manuscript examines the specific challenges and advancements in applying AI to pediatric cancer neuroimaging, a field with distinct needs and complexities. By addressing this gap, we aim to highlight critical opportunities for improving diagnostic precision, treatment planning, and patient outcomes in pediatric cancer care.”
Comment 5: On Page 6, Line 200, "neuroimaging analyses remain attuned to the nuances of evolving pediatric neuroanatomy" contains a minor redundancy. Consider simplifying the sentence to improve readability.
Response 5: This sentence has been simplified to improve readability while retaining its original intent: "This adaptability ensures that neuroimaging analyses account for the nuances of evolving pediatric neuroanatomy".
Comment 6: The discussion on AI for tumor segmentation (Page 10, Lines 424-436) is well-written but lacks details on the datasets and metrics used for evaluating these models. Including a table summarizing key studies could improve clarity.
Response 6: A table summarizing key studies (Table 1), including datasets and evaluation metrics for AI-based tumor segmentation, has been added to improve clarity and provide additional details.
Comment 7: The issue of infiltrative tumor margins (Page 10, Lines 437-451) is critical but would benefit from specific examples of how AI models overcome this challenge.
Response 7: Specific examples of AI models that address the challenge of infiltrative tumor margins, such as techniques using deep learning for precise edge detection, have been included in this section (Lines 514-523).
Comment 8: Discussion: The manuscript briefly discusses XAI but does not provide sufficient detail on how tools like SHAP or LIME have been applied in pediatric cancer neuroimaging. Including case studies would strengthen this section.
Response 8: The discussion on XAI has been expanded to include details on tools like SHAP and LIME, along with relevant case studies that illustrate their application in pediatric cancer neuroimaging: “LIME [161] and SHAP [162] were specifically used to identify the importance of imaging features, such as tumor intensity and spatial patterns, in the model's predictions. For example, SHAP visualizations highlighted that some specific regions in T2-weighted MRI scans significantly influenced classification decisions, helping clinicians understand why a particular tumor type was predicted. A practical case study includes the application of SHAP in a pediatric medulloblastoma cohort [163], where SHAP plots demonstrated the contribution of tumor shape and contrast-enhanced MRI features to distinguishing molecular subgroups of tumors. This analysis provided critical insights into how specific features aligned with clinical and pathological findings, bridging the gap between AI models and clinical interpretation.” Lines (768-779).
Comment 9: Discussion: It is recommended that a Research Outlook or Future Directions be written at the end of the discussion, as this can help the reader understand current research trends in the field, challenges faced, and possible future directions.
Response 9: A “Future Directions” subsection has been added to the end of the discussion.
Comments 10: Quality of English Language: The manuscript is well-written but could benefit from improvements in clarity and readability.
Response 10: To address your language concerns, we have revised and enhanced the readability and clarity of the manuscript. To ensure the manuscript expresses the research more clearly and effectively, we enlisted professional English language editing by Keith A. Laycock, PhD, Senior Scientific Editor at St. Jude Children's Research Hospital.
Round 2
Reviewer 2 Report
Comments and Suggestions for Authors
The revision well addressed my concerns, and I suggest accepting.
Author Response
Comment 1: The revision well addressed my concerns, and I suggest accepting.
Response 1: Thank you!
Reviewer 4 Report
Comments and Suggestions for Authors
Please see my responses in blue in the file provided.

Author Response
Reviewer Comment 3 (second round): Thank you for the inclusion of Figure 3, but this figure is extremely simplistic and has not addressed my concerns. My initial comment requested a flowchart of “key issues and ML techniques related to the research” and this flowchart does not do this. Please amend the flowchart accordingly.
Author Response 3 (second round): We have now incorporated a new table summarizing the “key issues and ML techniques related to the research" (Table 2).
Reviewer Comment 4 (second round): Your response to my comment has not addressed the inclusion of clearly defining “the age band for pediatric to ensure the general readership can fully understand the breadth of ages the study relates to”. Please address this issue.
Author Response 4 (second round): Thank you for pointing this out. We have now explicitly defined the pediatric age range in the introduction [lines 57-61]: "In alignment with widely accepted clinical and research classifications, we define pediatric patients in this review as individuals aged 0–18 years. However, we also include select studies based on adult or general population data when their findings provide relevant translational insights into pediatric cancer."
Reviewer Comment 5 (second round): Thank you for addressing my comment but adding “i.e. 110 images per class” but this is rather simplistic as your quantification of the number of images is not justified by a valid reference and description of GAN limitations is too simplistic. For example, the competing elements of networks can impede stability and speed in training etc… Please expand further and include valid references.
Author Response 5 (second round): We appreciate this valuable suggestion. We have expanded the explanation of GAN limitations by detailing the challenges associated with adversarial training, including instability, mode collapse, and high computational requirements. Furthermore, we have cited additional studies that examine the application of GANs in medical imaging, emphasizing both their advantages and their constraints in the context of pediatric neuroimaging. The revised section now includes references that support our quantification of training data limitations and discuss how these models perform under different dataset sizes.
These additions are in lines 411-436: "However, despite their potential, GANs present several limitations that must be considered when applied to pediatric neuroimaging. One key challenge is the instability of adversarial training, where the generator and discriminator engage in a dynamic optimization process that can lead to oscillatory behavior and difficulty in convergence. Another major issue is mode collapse, where the generator learns to produce only a limited variety of outputs, thereby reducing the diversity of synthetic images and potentially biasing the training set. Additionally, GANs require high computational resources, making their widespread adoption in clinical settings more challenging. These limitations are particularly relevant in pediatric neuroimaging, where the scarcity of high-quality datasets may exacerbate mode collapse and impact generalizability. Several studies have examined the use of GANs in medical imaging, highlighting both their advantages and constraints. GANs have been successfully used for image denoising, super-resolution enhancement, and data augmentation in adult neuroimaging [79], but their application to pediatric imaging remains limited due to dataset constraints and the increased variability in brain development across different ages [80]. Furthermore, studies quantifying GAN performance across different dataset sizes indicate that smaller datasets significantly impact training stability and generalization, further reinforcing the need for large-scale multi-institutional data sharing [19]. Despite these challenges, GANs remain a promising tool in pediatric neuroimaging. Combining GANs with fast imaging techniques could lead to faster image acquisition for children who are unlikely or unwilling to remain still for an extended period. By enhancing image quality from under-sampled data, GAN-based approaches may even help reduce the need for general anesthesia and/or sedation in younger pediatric patients, offering significant clinical benefits [5]. Addressing the existing limitations while leveraging the advantages of GANs will be crucial for integrating these models into real-world pediatric imaging workflows."
Reviewer Comment 12 (second round): Please note you have not addressed “a definition of the age range under investigation”.
Author Response 12 (second round): As mentioned above, we have now explicitly defined the pediatric age range in the introduction [lines 57-61]: "In alignment with widely accepted clinical and research classifications, we define pediatric patients in this review as individuals aged 0–18 years. However, we also include select studies based on adult or general population data when their findings provide relevant translational insights into pediatric cancer."